# Fast-Response Photodetector Based on Hybrid Bi_2_Te_3_/PbS Colloidal Quantum Dots

**DOI:** 10.3390/nano12183212

**Published:** 2022-09-16

**Authors:** Lijing Yu, Pin Tian, Libin Tang, Qun Hao, Kar Seng Teng, Hefu Zhong, Biao Yue, Haipeng Wang, Shunying Yan

**Affiliations:** 1School of Optics and Photonics, Beijing Institute of Technology, Beijing 100081, China; 2Kunming Institute of Physics, Kunming 650223, China; 3Yunnan Key Laboratory of Advanced Photoelectronic Materials & Devices, Kunming 650223, China; 4Department of Electronic and Electrical Engineering, Swansea University, Bay Campus, Fabian Way, Swansea SA1 8EN, UK; 5School of Materials and Energy, Yunnan University, Kunming 650500, China

**Keywords:** PbS, colloidal quantum dots, Bi_2_Te_3_, photodetector, hybrid

## Abstract

Colloidal quantum dots (CQDs) as photodetector materials have attracted much attention in recent years due to their tunable energy bands, low cost, and solution processability. However, their intrinsically low carrier mobility and three-dimensional (3D) confinement of charges are unsuitable for use in fast-response and highly sensitive photodetectors, hence greatly restricting their application in many fields. Currently, 3D topological insulators, such as bismuth telluride (Bi_2_Te_3_), have been employed in high-speed broadband photodetectors due to their narrow bulk bandgap, high carrier mobility, and strong light absorption. In this work, the advantages of topological insulators and CQDs were realized by developing a hybrid Bi_2_Te_3_/PbS CQDs photodetector that exhibited a maximum responsivity and detectivity of 18 A/W and 2.1 × 10^11^ Jones, respectively, with a rise time of 128 μs at 660 nm light illumination. The results indicate that such a photodetector has potential application in the field of fast-response and large-scale integrated optoelectronic devices.

## 1. Introduction

At present, HgCdTe (MCT), InSb, and type-II superlattices (T2SLs) are some of the widely used materials for infrared photodetectors [1]. These materials are usually grown under high-vacuum and high-temperature conditions using complex processes (such as epitaxial growth), hence resulting in high manufacturing cost. Furthermore, many existing photodetectors are required to operate in a relatively low-temperature environment in order to reduce the noise of the detection system, improve the sensitivity of detection, and reduce the influence of the thermal background on the performance of the device. The need for a cooling system would increase the overall size of the detection system and increase power consumption, as well as cost significantly, which often limits its application in the civilian market. Therefore, semiconductor materials that can be manufactured at low cost and exhibit excellent photodetection performance are highly desirable for the development of state-of-the-art photodetector technology.

Colloidal quantum dots (CQDs) have many significant advantages when they are used in photodetectors, for example, the optical and electrical properties can be adjusted by regulating the size and shape of the CQDs. Moreover, the nanomaterials can be solution-processed, and the device can be easily manufactured at low cost on almost any substrate materials [2]. To date, PbS CQDs are one of the most studied CQDs due to their well-established simple synthesis process. PbS CQDs have a large Bohr radius (18 nm) and a wide adjustable energy bandgap (0.6–1.6 eV). The first exciton peak of PbS can be adjusted from ultraviolet to short wavelength infrared. Therefore, PbS CQDs have become one of the most studied quantum dot materials for solar cells [3], ultraviolet and infrared photodetectors [4], and light-emitting diodes [5]. In addition, PbS CQDs demonstrated a high absorption coefficient (e.g., strong absorbability in visible and infrared regions) and good stability in air [6], hence the nanomaterial is ideal for the development of stable photoelectric devices and is a promising quantum dot material for optoelectronic applications [7].

However, the low carrier mobility (10^−5^–10^−2^ cm^2^·V^−1^·s^−1^) of PbS CQDs and numerous trap states in the nanomaterial will ease the recombination of the photogenerated carriers before they are being collected, which seriously affects the response speed and performance of the photodetector. Several methods have been reported to improve the performance of quantum dots. One of these methods was using ligand exchange to replace long insulating alkyl chains (e.g., oleic acid) during quantum dot synthesis to improve carrier mobility [8,9] and effectively reduce the defect density [10]. Another method was to combine quantum dots with other materials that exhibit high carrier mobility. This would result in a strong built-in potential, which can effectively improve the transport of carriers, response time and speed of the device [11]. Jeong et al. [12] reported a near-infrared photodetector based on a hybrid graphene/PbS CQD material, which exhibited a fivefold increase in the photocurrent, 22% increase in the rise rate, and 47% increase in the decay rate as compared with a PbS CQD device. In addition, a combination of transition metal disulfides (TMDs) and CQDs has also been reported. Kufer et al. [13] prepared a photoelectric transistor by combining MoS_2_ with PbS CQDs using MoS_2_ nanosheets as the electron transport layer. The responsivity of the device was higher by several orders of magnitude than photodetectors solely based on PbS CQDs and MoS_2_. Therefore, the combination of CQDs with two-dimensional (2D) materials that exhibit high carrier mobility can provide an effective solution to the slow response speed of CQD-based photodetectors. However, there are some limitations on hybrid 2D/CQD-based photodetectors due to the low light absorption of the 2D materials resulting in a low response rate over a broadband. The use of 3D topological materials can potentially provide a new solution to the low absorption of 2D materials. Since the discovery of the quantum Hall effect, 3D topological insulator materials, such as Bi_2_Te_3_, have attracted much attention due to their unique energy bandgap structure (e.g., an insulating energy gap in the bulk with gapless edge or surface states) [14]. Bi_2_Te_3_ has been widely used in the study of broadband photoelectric detection due to its narrow bulk bandgap (0.17 eV) and high carrier mobility (e.g., surface carrier mobility of 5800 cm^2^·V^−1^·s^−1^) without external influence [15]. In 2016, Wang et al. [16] reported a photovoltaic detector consisting of n-type topological insulator Bi_2_Te_3_ thin films grown on p-type silicon substrates, and the device demonstrated good photovoltaic effect over a broadband range from ultraviolet (UV) to near-infrared (NIR). A short-circuit current of 19.2 μA and an open-circuit voltage of 235 mV were achieved under 1000 nm illumination. By taking advantage of the strong bulk bandgap optical absorption and high surface carrier mobility of 3D topological materials, the combination of 3D topological materials with CQDs can offer an excellent solution to the slow response speed of CQDs as well as the low response of 2D materials under a broad spectrum. Presently, there is limited report on photodetectors based on hybrid 3D topological insulating materials and CQDs. Most of the work on 3D topological materials is concerned with the quantum spin Hall effect, and little attention has been paid to its application in the field of photodetectors.

In this paper, high-quality Bi_2_Te_3_ thin films and PbS CQDs with a uniform size distribution were prepared. A heterojunction photodetector consisting of hybrid Bi_2_Te_3_/PbS CQDs in a device structure of indium tin oxide (ITO)/ [6,6]-phenyl-C61-butyric acid methyl ester (PCBM)/PbS/Bi_2_Te_3_/Al was developed and studied in which the advantages of topological insulators and CQDs were realized. The responsivity (*R*) and detectivity (*D**) of the device were 18 A/W and 2.1 × 10^11^ Jones, respectively, with a fast response time of 128 μs.

## 2. Materials and Methods

### 2.1. Materials

ITO grown on a quartz substrate was purchased from Beijing Jinji Aomeng Technology Co., Ltd., Beijing, China. PbS CQDs were synthesized by thermal injection method as reported by Hines Ma et al. [17]. The as-prepared PbS CQDs were dissolved in an n-octane solvent with a concentration of 30 mg/mL (n-octane was purchased from Tianjin Zhiyuan Chemical Reagent Co., Ltd., Tianjin, China). PCBM was dissolved in chloroform with a concentration of 100 mg/mL (PCBM and chloroform were purchased from Jilin OLED Material Tech Co., Ltd., Changchun, China. and Chengdu Chron Chemicals Co., Ltd., Chengdu, China, respectively). The Bi_2_Te_3_ film was deposited using a magnetron-sputtering technique. Electrical contact pads consisting of Al electrodes were evaporated in a vacuum metal evaporator. Bi_2_Te_3_ targets (99.99%) and Al slice (99.99%) were all purchased from Zhongnuo Advanced Material (Beijing) Technology Co., Ltd., Beijing, China.

### 2.2. Device Fabrication

After cleaning and drying the ITO substrate, a layer of PCBM was spin-coated onto the substrate at a rotational speed of 2500 rpm for a duration of 30 s. Subsequently, a solution of PbS CQDs was spin-coated on the PCBM film. Tetrabutyl-ammonium iodide (TBAI) was introduced on the PbS CQD layer and rested for 60 s before spin-coating. The duration of spin-coating was set at 30 s for each step with a rotational speed of 2500 rpm. The coated film was then rinsed using methanol. Ten layers of PbS CQDs films were spin-coated using the same method. This was followed by the deposition of the Bi_2_Te_3_ film by magnetron-sputtering at an Ar flow of 60 standard cubic centimeters per minute (sccm). The sputtering was carried out at room temperature with a sputtering power of 200 W, sputtering pressure of 5 Pa, and sputtering duration of 1 s. Finally, Al electrodes were evaporated in a vacuum metal evaporator. The magnetron sputtering equipment and vacuum metal evaporator were all purchased from Shenyang Kecheng Vacuum Tech Co., Ltd., Shenyang, China.

## 3. Results and Discussion

PbS CQDs exhibited strong absorption from ultraviolet to near infrared (the UV-Vis absorption spectrum of PbS CQDs was shown in Appendix A), they were used as an absorption layer and photoelectric conversion layer in the device. The morphology and size distribution of the PbS CQDs were investigated by transmission electron microscopy (TEM). As presented in Figure 1a, the as-synthesized PbS CQDs showed excellent monodispersity with an average particle size of 4.01 nm and a full width at half maximum (FWHM) of 0.49 nm. High-resolution TEM (HRTEM) was performed to study the lattice fringes of PbS CQDs. As shown in Figure 1b, three different lattice structures were observed with d-spacings of 0.213, 0.340, and 0.293 nm, which corresponded to (220), (111), and (200) crystal planes of PbS CQDs, respectively. An angle of 54.5° was measured between (111) and (200), which is in agreement with the theoretical calculation of 54.7°. Figure 1c shows the line profiles of the three lattice fringes in Figure 1b as denoted by three different colors. The line profiles from top to bottom corresponded to the (220), (111), and (200) crystal planes, respectively. Schematic diagrams illustrating the crystal structure of PbS CQDs along the (220), (111), and (200) crystal planes are shown in Figure 1d–f, respectively.

Structural characterization and analysis on the 3D topological insulator material Bi_2_Te_3_ were performed as it is an important functional layer in the device. Figure 1g shows a low-resolution TEM image of the Bi_2_Te_3_ film. An area on the image in Figure 1g was selected to study the crystal microstructure at high resolution as shown in Figure 1h. Crystal lattice spacing of 0.314 nm was measured that corresponded to the (015) crystal plane of Bi_2_Te_3_. The selected area electron diffraction (SAED) pattern of the Bi_2_Te_3_ film is shown in Figure 1i.

The thickness of the Bi_2_Te_3_ film has a significant effect on the performance of the photodetector as it can influence the optical absorbance of the film as well as the diffusion length of the carriers. Figure 2a shows a Bi_2_Te_3_ film with a thickness of 7.5 nm deposited on a silicon dioxide substrate using the same process conditions as in the device fabrication. The internal molecular vibration state of the material was studied by Raman spectroscopy to determine the phase of the as-prepared film. The crystal structure of Bi_2_Te_3_ is in the space group R3m having a layered structure in the order of -Bi-Te(1)]-[Te(1)-Bi-Te(2)-Bi-Te(1)]-[Te(1)-Bi- [18]. Bi_2_Te_3_ has 15 vibrational modes, and the optical modes that can be detected by Raman spectroscopy are E_g_, A_1g_, E_u_, and A_1u_ [19]. E_g_ is generated by the in-plane vibration of the five-layer structure, and A_g_ is generated by the out-of-plane vibration of the five-layer structure. Furthermore, A_1u_ is due to five-layer defects [20]. At T = 300 k, there are four Raman active lattice vibrations with wavenumbers at 36.5 cm^−1^ (E^1^_g_), 62.0 cm^−1^ (A^1^_1g_), 102.3 cm^−1^ (E^2^_g_), and 134.0 cm^−1^ (A^2^_1g_) [18]. As shown in Figure 2b, three vibrational peaks at 62.6 cm^−1^ (A^1^_1g_), 101.1 cm^−1^ (E^2^_g_), and 130.7 cm^−1^ (A^2^_1g_) were observed, and they are in good agreement with previously reported work, therefore suggesting a successful preparation of the Bi_2_Te_3_ film. Figure 2c depicts the vibrational mode of Bi_2_Te_3_ [21]. In the A^1^_1g_ vibration mode, Bi and Te(1) were vibrating in phase. However, in the A^2^_1g_ mode, Bi and Te(1) were vibrating out of phase. The adjacent Te(1) atoms always vibrate out of phase [18].

Figure 2d shows the XRD pattern of the Bi_2_Te_3_ film with characteristic peaks located at 17.7°, 27.6°, 38.1°, 54.8°, and 62.4°, corresponding to the crystal planes of (006), (015), (1010), (1016), and (1115), respectively (according to the standard card pdf# 080027). This is in good agreement with previously reported work [22], which implied that the prepared Bi_2_Te_3_ film exhibited good crystalline quality. Xray photoelectron spectroscopy (XPS) was used to analyze the elemental composition and surface oxidation state of the Bi_2_Te_3_ film. The Bi 4f core level (as shown in Figure 2e) consisted of peaks at 164.3, 163.1, 159, and 157.8 eV, corresponding to Bi 4f_7/2_ (oxide), Bi 4f_5/2_ (metal), Bi 4f_5/2_ (oxide), and Bi 4f_7/2_ (metal), respectively [23]. Figure 2f shows the core level peaks of Te 3d situated at 586.3, 582.6, 575.9, and 572.2 eV, which corresponded to Te 3d_3/2_ (oxide), Te 3d_3/2_ (metal), Te 3d_5/2_ (oxide), and Te 3d_5/2_ (metal), respectively, similar to previously reported work [24,25]. The above studies showed that the Bi_2_Te_3_ film was deposited but exhibited some degree of surface oxidation due to its interaction with ambient air. Interestingly, Bi_2_Te_3_ (as a 3D topological insulating material) has strong surface states, which are unlikely to be influenced by its oxidation states; hence, they will have little effect on the performance of the device.

The fabrication process of the photovoltaic detector with the device structure of ITO/PCBM/PbS/Bi_2_Te_3_/Al is illustrated in Figure 3a as described in Section 2.2. Figure 3b shows a cross-sectional scanning electron microscope (SEM) image of the device structure. The thicknesses of ITO, PCBM, PbS, Bi_2_Te_3_, and Al were approximately 270, 77, 150, 7, and 95 nm, respectively.

The mechanism and advantages of this device structure can be explained using the energy band diagram as depicted in the inset of Figure 3c. Electrons and holes are generated at the PbS photosensitive layer. The electrons are transported to the ITO electrode through the PCBM layer, and the holes are transported to the Al electrode through the Bi_2_Te_3_ layer, hence forming photogenerated carriers. Due to the higher conduction band minimum position of PbS CQDs, electrons are prevented from entering the hole transport layer via the PCBM layer, effectively reducing the carrier loss due to recombination. The valence band maximum of PbS CQDs is lower than the highest occupied molecular orbital (HOMO) of Bi_2_Te_3_, which reduces the potential barrier for hole transport and thus facilitates hole transport from the PbS CQD layer to Bi_2_Te_3_ layer. Therefore, the hybrid layer structure is beneficial to the transport of carriers and the collection of photogenerated carriers, which will lead to significant improvement on the performance of the device.

The performance of the photodetector was characterized and analyzed. Since the devices were capable of responding to wavelength from UV to visible bands (*I–V* plots under dark and light illumination at 365, 500 and 850 nm were shown in Appendix A), we had chosen an intermediate band (e.g., 660 nm) for the detailed studies. *I–V* measurement was performed on the device using Keithley 2400 under dark and light illumination at 660 nm with power density of 2380 μW·cm^−2^. Figure 3c shows the *I–V* characteristic curves of the device. It is evident that the device produced photogenerated current under light illumination. The responsivity (*R*) and detectivity (*D**) of the device can be calculated using the following equations [26]:*R*(*λ*) = *J*_ph_(*λ*)/*P*_opt_(*λ*)(1)
*D**(*λ*) = *R*(*λ*)/(2*qJ_dark_*)^1/2^(2)
where *λ* is the incident light wavelength, *J*_ph_(*λ*) is the photocurrent density, *P*_opt_(*λ*) is the optical power density at a specific wavelength, *q* is the unit charge, and *J*_dark_ is the dark current density. All measurements were performed at room temperature. The responsivity and detectivity obtained by the above equations are shown in Figure 3f. The maximum responsivity and detectivity at 660 nm were 18 A/W and 2.1 × 10^11^ Jones, respectively. The photoelectric response time of the device is shown in Figure 3d,e. The device was stored up to 6 months and characterized after storage of 1, 3, and 6 months. The results showed that the device performance was stable and reproducible under 660 nm illumination after storage of 6 months. The rise and fall times were 128 µs and 3 ms, respectively. Table 1 provides a list of other photodetectors based on PbS QD heterojunctions and their performances. It shows that the device from this work demonstrated excellent response time as compared with other devices.

## 4. Conclusions

In this paper, a fast-response high-performance photodetector based on hybrid PbS CQDs/Bi_2_Te_3_ was prepared. Combining the synergistic effect of PbS CQDs and a Bi_2_Te_3_ film, the photodetector exhibited a fast response time of 128 μs with responsivity and detectivity of 18 AW^−1^ and 2.1 × 10^11^ Jones, respectively. The response time of the device was faster by several orders of magnitude than that of photodetectors consisting of solely PbS CQDs and other PbS QD heterojunctions. The energy band structure of the device is beneficial to the fast carrier transport and collection of photogenerated carriers, thereby improving the response rate and performance of the device. With the advantages of excellent performance, low cost, and solution processability, the hybrid PbS CQDs/Bi_2_Te_3_ photodetector has promising applications in the field of fast-response and large-scale integrated QD-based optoelectronic devices.

## Figures and Tables

**Figure 1 nanomaterials-12-03212-f001:**
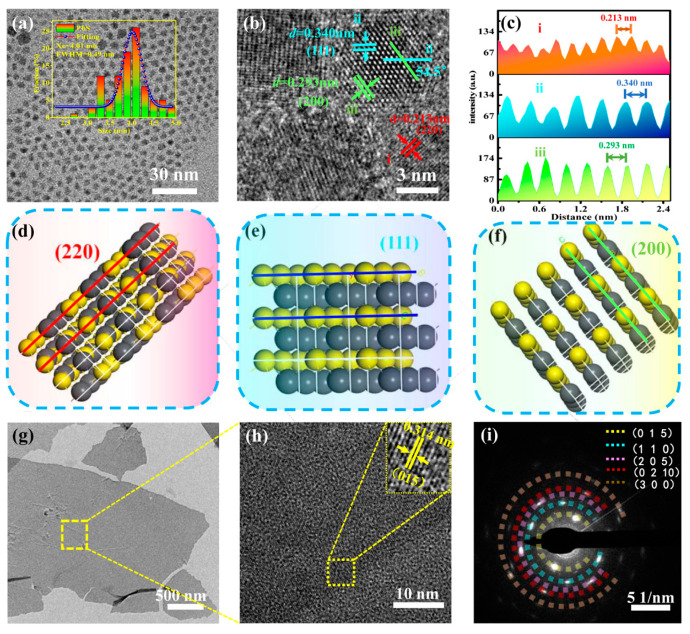
Structural characterization of PbS CQDs and Bi_2_Te_3_ film. (**a**) Low-resolution TEM image of PbS CQDs. (**b**) HRTEM image of PbS CQDs. (**c**) Line profiles corresponding to the lines i–iii indicated in (**b**). (**d**–**f**) Schematic diagrams of PbS CQD crystal structures. (**g**) Low-resolution TEM image of Bi_2_Te_3_ film. (**h**) HRTEM image showing lattice fringes of Bi_2_Te film. (**i**) Electron diffraction pattern of Bi_2_Te_3_ film.

**Figure 2 nanomaterials-12-03212-f002:**
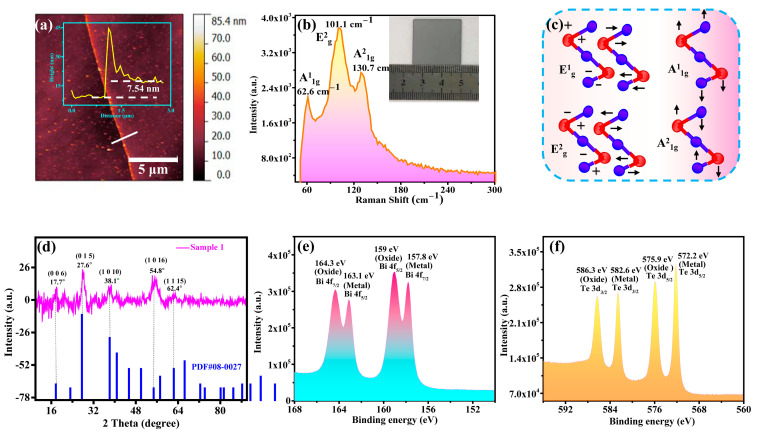
Structural characterization and phase analysis of Bi_2_Te_3_ film. (**a**) AFM image of the film and line profile (inset) for film thickness measurement. (**b**) Raman spectroscopy of the film. (**c**) Schematic diagram illustrating displacement patterns of phonons. (**d**) XRD pattern of the film. (**e**,**f**) XPS spectra of Bi 4f and Te 3d core levels of the film.

**Figure 3 nanomaterials-12-03212-f003:**
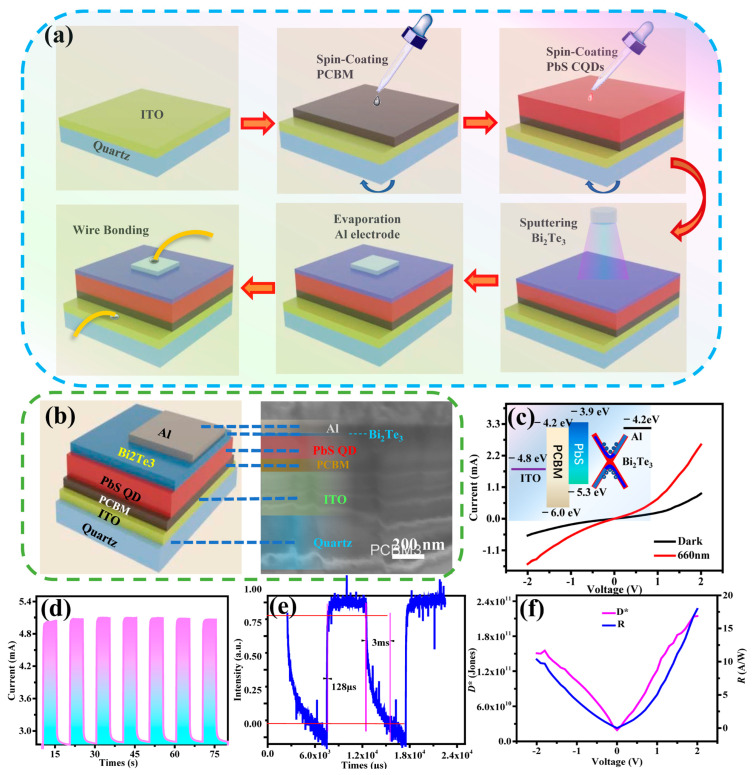
(**a**) Schematic diagrams illustrating fabrication process of the photodetector. (**b**) Schematic diagram and crosssectional SEM image of the photodetector. (**c**) *I–V* measurements under dark and light illumination at 660 nm (inset: energy band structure of the device). (**d**,**e**) Photocurrent switching behaviors. (**f**) Plot of responsivity (*R*) and detectivity (*D**) against voltage (V).

**Table 1 nanomaterials-12-03212-t001:** List of photodetectors based on PbS QD heterojunctions and their performances.

Device	Excitation Wavelength (nm)	Rise/Fall Time	Responsivity (A/W)	Detectivity (Jones)	Ref.
PCBM/PbS-TBAl/Bi_2_Te_3_	660	128 (μs)/3 (ms)	18	2.1 × 10^11^	This work
PbS-TBAI/SiNx/Si	1064/1310	160/320 (μs)	0.68/0.29	7.74 × 10^10^/3.32 × 10^10^	[27]
ZnO/PbS-TBAl/PbS-EDT	500/910	25.5/25.6 (ms)	385/444	3.9 × 10^13^/4.52 × 10^13^	[28]
PbS-QD/graphene	1550	3/200 (ms)	104	10^12^	[29]
PbS-QD/graphene	635/1600	-/200 (ms)	-	4 × 10^12^	[30]
PbS-QD/WSe_2_	970	7/480 (ms)	2 × 10^5^	7 × 10^13^	[31]
PbS-QD/MoS_2_	400–1500	-/0.3 (s)	6 × 10^5^	5 × 10^11^	[13]

## Data Availability

All data, models, and codes generated or used during the study appear in the submitted article.

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
