# Peer review of "Fast-Response Photodetector Based on Hybrid Bi2Te3/PbS Colloidal Quantum Dots"

_nanomaterials, 2022, doi:10.3390/nano12183212_

Round 1

Reviewer 1 Report

In their manuscript, the authors propose a new hybrid photodetector which combines the advantages of PbS colloidal quantum dots (CQDs) and the 3D topological insulator bismuth telluride (Bi2Te3). As a result, the authors are able to fabricate a device with a maximum responsivity of 18 A/W and a detectivity of 2.1×10^11 Jones, with a rise time of 128 μs at 660 nm light illumination. The structural characterisation of PbS CQDs and Bi2Te3 films is carried out by combining TEM, AFM, XPS, Raman and XRD analysis. The optoelectronic investigation of the photodetector properties is carried out via IV measurements under dark and light illumination.

The overall work seems solid and the conclusions the authors draw are backed by the experimental data. Also, the paper is clearly written. I have a few comments for the authors which I summarise in the following.

1) What are the capping ligands of the as-prepared CdSe CQDs? Do they play any role in device performances? How did the authors choose the ligands?

2) I would expect to see at least a UV-vis absorption spectrum of the active layer of the photodetector together with its discussion to better understand the optical properties of the device.

3) Why are device performances measured at 660nm? The authors should explain their choice. Also, what happens at other wavelengths?

4) The authors should provide information on the number of devices they have fabricated to measure the IV curves and the statistical error on the responsivity (R) and detectivity (D*).

5) The authors should clarify what they mean when they say “The device exhibited good stability under 660 nm illumination.” “Good” without a number does not mean anything.

6) In Table 1, it is not clear to me whether the numbers in the rise/fall time column refer to the different excitation wavelengths or to the rise and fall times. Also, as far as I understood, in the table, there is no fall time for  PCBM/PbS-TBAl/Bi2Te3 and maybe also for the other samples.

After the authors have successfully addressed these points, the manuscript may be suitable for publication in Nanomaterials.

Reviewer 2 Report

REVIEW

Fast response photodetector based on hybrid Bi2Te3/PbS colloidal quantum dots

by Lijing Yu, Pin Tian, Libin Tang, Qun Hao, Kar Seng Teng, Hefu Zhong, Biao Yue, Haipeng Wang, and Shunying Yan 

In the presented manuscript the results of experimental investigations of the photoelectrical properties of photodetectors based on multilayered structures with PbS colloidal quantum dots and Bi2Te3 3D topological insulator are presented. Transmission electron microscopy, scanning electron microscopy, selected area electron diffraction, Raman spectroscopy, and X-ray diffraction methods are applied to characterize the structural quality of the obtained structures. Current-voltage characteristics under dark and light illumination, time-dependent photocurrent, photoresponsivity, and detectivity of the proposed photodetectors are analyzed. Comparison with some similar photodetectors is carried out. Rather low response time is obtained for the proposed photosensitive structures. These structures and technological approach to the device design should be useful for further device applications in optoelectronics.

Chosen methods are adequate and results of experimental research are reliable. Overall, the paper is organized and written very well and at high scientific level.

The main recommendation about this manuscript is to add more detailed description of layers (especially those with quantum dots) in the investigated structures.

Moreover, in the reviewer’s opinion, the manuscript should be slightly improved before publication. The details are listed below.

1.      Page 2, Introduction: ITO/PCBM abbreviations should be deciphered or removed from this section.

2.      Page 3, Introduction: 128 us (microseconds) should be written using Greek letter mu.

3.      Page 3, Device fabrication section: on to preposition should be without spacing.

4.      Page 3, Device fabrication section: sccm abbreviation should be deciphered.

5.      Page 3, Results and discussion section: Information on estimated concentration (surface density) of colloidal quantum dots and the thicknesses of these layers is recommended to be added.

6.      Page 4, Results and discussion section: Te1-Bi-Te2-Bi-Tel should be checked.

7.      Page 4, Results and discussion section: Commas should be checked in the list of modes and wavenumbers.

8.      Page 4, Results and discussion section: The following sentence “Bi and Te(1) vibrate in phase as denoted by A11g vibrational mode, while Bi and Te(1) vibrate out of phase as denoted by A21g vibrational mode” seems to be incorrect.

9.      Page 4, Results and discussion section: Bi 4f7/2 (oxide), Bi 4f5/2 (metal), Bi 4f7 /2 (oxide) and Bi 4f5/2 (metal) phrase should checked to avoid duplication.

10.  Page 7, Results and discussion section: What temperature was used for measurements of current-voltage characteristics under dark and light illumination?

11.  Table 1: Information on the fall time of the photodetectors considered in the presented manuscript is missed.

Conclusion: The presented manuscript may be published in the Nanomaterials journal after minor revision.

Reviewer 3 Report

The authors have written a very interesting and extremely useful paper on the creation of new photodetectors based on the combination of useful properties of topological insulator and colloidal quantum dots. Namely, hybrid BiTe3/PbS photodetector. This photodetector demonstrated maximum responsitivity and detectivity with rise time of 128 microseconds at wavelength of 660 nm. This response time is several orders of magnitude higher than that of photodetectors made of solely colloidal quantum dots. In fact, perhaps we are talking about a new direction in the construction of new types of fast response photodetectors. I think the paper can be published in present form.

Round 2

Reviewer 1 Report

I believe the authors have addressed adequately all the comments in previous referee reports and, for this reason, I recommend the acceptance of the paper for publication in Nanomaterials.